# Variability of the lymphocyte-to-monocyte ratio in patients with chronic kidney disease on hemodialysis

Daysi Zulema Diaz-Obregón[1], Gabriela Goyoneche Linares[2], Ana Granda Alacote[1],
Alexis G. Murillo Carrasco[3]*, Michael Bryant Castro Núñez[4],
Andrea Vanessa Llanos Diaz[2], Katherine Susan Rufasto Goche[5],
Antonio Mauricio Sánchez Cotrina[6], Víctor Arrunátegui Correa[1†], Joel de León Delgado[1]

1 Universidad de San Martín de Porres, Facultad de Medicina Humana, Lima, Peru, 2 ONG INNOVACARE, Lima, Perú, 3 Organization for Medical Innovation and Collaboration for Sciences—OMICS, Lima, Peru, 4 Escuela de Posgrado, Universidad Nacional Mayor de San Marcos, Lima, Perú, 5 Facultad de Odontología, Universidad Nacional Federico Villarreal, Lima, Peru, 6 Universidad de Piura, Lima, Peru

† Deceased.
* agmurilloc@gmail.com

## Abstract

### Introduction

Chronic kidney disease (CKD) is a global public health issue characterized by a state of persistent inflammation that leads to immune system dysregulation. The lymphocyte-to-monocyte ratio (LMR) has emerged as an inflammatory biomarker of interest for monitoring the progression of this condition.

### Objective

To determine the clinical-epidemiological, biochemical, and hematological characteristics associated with differences in the LMR in patients with chronic kidney disease undergoing hemodialysis.

### Materials and Methods

A retrospective cohort study was conducted by reviewing the medical records of 120 CKD patients on hemodialysis treated at a private nephrology center in Lima, Peru. Patients were divided into two groups based on the median LMR value: high LMR (< 3.06) and low LMR (≥ 3.06). Logistic regression was used to analyze risk factors associated with LMR variation, complemented by longitudinal regression analysis (up to three years follow-up).

### Results

The median age of the patients was 58 years at the time of analysis. Chronic glomerulonephritis, arterial hypertension, and type 2 diabetes mellitus (T2DM) were

**Data availability statement:** All relevant data are within the paper and its Supporting Information files.

**Funding:** The author(s) received no specific funding for this work.

**Competing interests:** The authors have declared that no competing interests exist.

the most frequently reported causes of end-stage CKD. A total of 54.2% of patients had been on hemodialysis for more than seven years. Risk factors associated with decreased LMR, and thus with increased inflammation, were: older age (OR = 1.03, $p < 0.001$), more than three years on hemodialysis (OR = 2.17, $p = 0.002$), more than seven years on hemodialysis (OR = 2.38, $p < 0.001$), and presence of T2DM (OR = 2.2, $p = 0.006$). In addition, a direct positive contribution was found with hematocrit levels (Beta = 0.25, $p < 0.001$) and the presence of an arteriovenous fistula (Beta = 0.17, $p = 0.038$).

## Conclusions

A reduced LMR is a useful inflammatory biomarker for monitoring CKD severity. Increased age, prolonged duration of hemodialysis (more than three years), presence of T2DM, and elevated ferritin levels are associated with lower LMR values. In contrast, higher hematocrit levels and the presence of an arteriovenous fistula are directly associated with higher LMR values.

## Introduction

Chronic kidney disease (CKD) is frequently accompanied by a persistent low-grade inflammatory state, which plays a role in disease progression and is linked to a spectrum of comorbid conditions including atherosclerosis, cardiovascular diseases, cachexia, malnutrition, and anemia [1]. This condition affects more than 10% of the global population [1]. In 2016, it ranked 16th among the leading causes of death and is projected to rise to fifth place by 2040 [2].

The prevalence of CKD in Peru is 13.2% [3]. However, in the capital city, Lima, home to one-third of the country's population, it has been estimated at 20.7%. CKD is the sixth leading cause of death in the country, with a 28.0% increase over the past 10 years [3].

Diabetes mellitus (DM) and hypertension are associated with the progression of CKD to end-stage renal disease (ESRD) requiring renal replacement therapies in about 70% of cases [4]. However, both DM and hypertension are modifiable risk factors [5].

On an immunological level, CKD patients undergoing hemodialysis therapy show persistent immune system activation, characterized by the expansion of circulating monocytes, decreased phagocytic and antigen-presenting capacities, and T and B lymphocyte dysfunction. This immune profile is associated with sustained release of pro-inflammatory cytokines such as IL-1, IL-6, IL-12, TNF-α, and reactive oxygen species, among others, which induce a chronic systemic inflammatory state marked by high oxidative stress and endothelial damage and dysfunction [6].

Various studies have confirmed that inflammation plays a critical role in CKD progression. In particular, IL-6 contributes to vascular and systemic endothelial damage by reducing the expression of endothelial nitric oxide synthase, the enzyme responsible for nitric oxide production in endothelial cells, which is crucial for vasodilation, inhibition of platelet aggregation, and protection against vascular inflammation [6].

TNF-α alters renal hemodynamics, promotes immune cell infiltration, and stimulates apoptosis. Although IL-10 is traditionally considered anti-inflammatory, it may have a dual role. In chronic inflammatory contexts such as CKD, its dysregulated expression may contribute to immune dysfunction by negatively modulating antigen-presenting cell responses and altering immune homeostasis. Moreover, type I interferons (IFN-α, IFN-β) have been associated with worsening renal damage, while IFN-γ (type II) has a pro-inflammatory role that accelerates CKD progression [6,7].

Regarding lymphocyte-related indices, the lymphocyte-to-monocyte ratio (LMR) and its inverse, the monocyte-to-lymphocyte ratio (MLR), have been linked to inflammatory processes. Variability in lymphocyte and monocyte populations has been associated with both acute and chronic diseases [8–10]. These associations are grounded in the presence of distinct monocyte subpopulations: "intermediate" (CD14++CD16+) and "non-classical" (CD14+CD16++), both of which are proinflammatory, unlike "classical" monocytes (CD14++CD16–), which are primarily phagocytic and involved in immune defense [11].

In CKD, variations in monocyte and lymphocyte lineages have also been described. For example, during hemodialysis, a reduction in intermediate monocytes is associated with a decrease in cardiovascular events and lower mortality; meanwhile, in patients with DM, glycemic variability is linked to increased expression of intermediate monocytes and higher cardiovascular risk even during transient hyperglycemia episodes [12,13].

In CKD, chronic inflammation and immune dysfunction play a key role in the progression of cardiovascular damage. In this context, intermediate monocytes have gained attention due to their pro-inflammatory phenotype and distinctive expression of chemokine receptors related to atherosclerosis [14]. Several studies have shown that higher counts of this monocyte subpopulation are associated with renal function decline, and increased incidence of cardiovascular events, as well as higher mortality in dialysis patients and those in early CKD stages [9,15]. Low LMR values are also associated with inflammation in other chronic diseases such as esophageal cancer, pancreatic cancer, and leukemia [16–18], aortic stenosis progression [19], and prediction of peripheral artery disease [20]. Conversely, elevated MLR values, reflecting lower LMR, have been linked to aging [21], increased cardiovascular risk [22], CKD progression [10,23], and high mortality due to CKD [22].

In patients with type 2 diabetes mellitus (T2DM) and CKD treated in intensive care units, LMR has been shown to significantly correlate with all-cause mortality risk at 90 days [24]. Zhou et al. (2024) demonstrated that MLR is a simple and low-cost biomarker that reflects systemic inflammation and helps physicians in the management of nephropathies [10]. In CKD patients on hemodialysis, CD16+monocytes increase, which is associated with higher risk of progression to endstage disease [13,15,25].

Recently, the importance of monocyte heterogeneity has emerged, with specific monocyte subpopulations increasing in CKD and also serving as predictors of cardiovascular disease in this population [26].

Evidence for LMR as a predictor of disease severity was described by Wang (2017), who found an inverse relationship between intermediate monocyte count and the development of cerebrovascular events and severity of calcified aortic stenosis [27]. These findings, although promising, remain insufficient, as no clinically validated cut-off points for LMR or MLR have been established as biomarkers of inflammation in CKD patients, highlighting the need for further investigation.

T2DM is one of the main etiological causes of CKD and is characterized by a highly inflammatory and oxidative state. In this context, LMR is particularly useful for monitoring patients with diabetic nephropathy, as it reflects the progression of the inflammation characteristic of this condition. This may help explain the high morbidity and mortality observed in T2DM patients [6,12,15,25,28].

Monocyte count has been identified as an indirect marker of systemic inflammation, reflecting elevated levels of circulating pro-inflammatory mediators and being linked to a higher risk of cardiovascular disease (CVD). In this context, LMR has gained attention as a useful hematological index for predicting cardiovascular risk. While monocytes are actively involved in the inflammatory response, lymphocytes (including T, B, and Natural Killer cells) play a key role in immune regulation [22].

Given this evidence, there is growing interest in LMR (and MLR) as biomarkers of inflammation and chronic disease severity. To date, its predictive value has been reported in chronic diseases such as peripheral arterial disease, peripheral neuropathy, osteomyelitis, and cancer, among others [20,29].

Therefore, it is pertinent to analyze the lymphocyte-to-monocyte ratio (LMR) and its inverse, MLR, as a biomarker of inflammation and progression of chronic kidney disease (CKD), due to its potential clinical utility, low cost, and ease of measurement. The available evidence on its application in patients with chronic nephropathy is still limited, and there are significant gaps regarding clinically validated cut-off points. For this reason, we conducted this study to analyze the socio-demographic, clinical, biochemical, and hematological characteristics associated with variations in LMR in Peruvian CKD patients on hemodialysis treated at a private nephrology center during the period 2018–2020.

## Methods

### Study design and population

This was an observational, analytical, retrospective cohort study, conducted at the private nephrology center CENESA S.A., located in Lima, Peru. This center receives patients from the Edgardo Rebagliati Martins National Hospital (Peruvian Social Health Insurance – ESSALUD). The study population included 162 ESRD patients who underwent hemodialysis therapy between January 2018 and December 2020. Follow-up ranged from 6 to 36 months. A census sampling approach was used. Data was collected from clinical records on December 10, 2020.

### Variables and measurements

Data were collected from medical records and daily progress notes of patients who met the inclusion criteria. The dependent variable was the LMR, while the independent variables included sociodemographic characteristics (sex, age), body mass index (BMI), etiology of ESRD, duration of hemodialysis, type of vascular access, and laboratory parameters (hematologic and biochemical tests). Laboratory test results from pre-dialysis and the third week of each month were analyzed.

### Inclusion and exclusion criteria

All patients over 18 years of age with ESRD and a minimum of six months on hemodialysis were included. A total of 42 patients were excluded: 15 hospitalized for infections, trauma, cardiovascular and cerebrovascular emergencies (4 with T2DM); 11 deceased (1 with diabetes); 10 with less than six months on hemodialysis; 4 with autoimmune diseases; and 2 with oncological conditions. The 120 patients who met the inclusion criteria were divided into two groups: low LMR patients (n1 = 61) and high LMR patients (n2 = 59).

### Sensitivity analysis

To determine the most informative threshold values for the lymphocyte-to-monocyte ratio (LMR), a sensitivity analysis was conducted focusing on variables that previously showed significant associations in the stratified group analysis (based on high vs. low LMR levels). Specifically, six variables were selected: Age, Time on Hemodialysis (HD), Monocyte count, Lymphocyte count, Neutrophil-to-Lymphocyte Ratio (NLR), and Platelet-to-Lymphocyte Ratio (PLR).

For each variable, the dataset was dichotomized based on the median value. Receiver Operating Characteristic (ROC) curves were then constructed to evaluate the ability of various LMR thresholds to discriminate between the two subgroups. Using the pROC package in R, we identified the three LMR cutoff points with the optimal balance of sensitivity and specificity. These points were selected based on the Youden Index, and graphically annotated on each curve. The area under the ROC curve (AUC) was calculated to quantify the discriminative performance of LMR about each variable.

## Statistical analysis

Data were recorded in a database using Stata version 17 and R v.4.4.2. Descriptive tests were performed for all variables (95% CI). For bivariable analysis, chi-square or Fisher's exact test, Student's t-test or Mann–Whitney U test, and ANOVA or Kruskal–Wallis' test was used. For multivariable analysis, a logistic regression model was applied to estimate the Odds Ratio (OR). LMR was categorized as "high" and "low" using the population median of 3.06. Mixed linear models were used to assess the relationship between the dependent variable LMR and a set of clinical predictors, accounting for potential clustering of data. Models were fitted using the lmer() function from the *lme4* package in R v4.4.2. Fixed effects included age, vascular condition, hematocrit, and hemoglobin, and a random intercept for each patient was added to model individual-level variability.

## Ethical considerations

This study was approved by the Ethics Committee of the Universidad de San Martín de Porres under Official Letter No. 473–2019-CIEI-FMH-USMP, issued on July 23, 2019. As the study did not involve any patient interventions, informed consent was waived. There was no direct contact with patients, and data confidentiality was strictly maintained during data collection (December 10, 2020).

## Results

### Population characteristics

The median age of the population was 58 years (47–72), with a predominance of males (54.2%). A total of 65.8% of patients had a normal BMI. The most frequent cause of ESRD was chronic glomerulonephritis (34.2%), followed by hypertension (25.8%) and T2DM (23.3%). About 54.2% had been on hemodialysis for more than seven years, and 65% of patients had an arteriovenous fistula (AVF) as vascular access (Table 1). Our dataset is provided as the S1 File.

### Comparative analysis based on the Lymphocyte-to-Monocyte Ratio (LMR)

To compare patients with low and high LMR levels, the population baseline median (3.06) was used as the cut-off point. The Fig 1 shows the full landscape of LMR levels in this Peruvian population with a maximum follow-up time of 35.5 months. A significant association was found between low LMR and increased age (p = 0.01) and less time on hemodialysis (p = 0.02). No significant differences were found in other clinical-epidemiological characteristics (Table 2), and neither LMR (p = 0.7) nor sex (p = 8) was influenced by age in this study.

At the hematological level, all patients with low LMR presented lymphopenia, with a highly significant difference compared to those with high LMR (p < 0.001). Monocyte count was higher in the low LMR group (p < 0.001), also with a highly significant difference, although monocyte counts in both groups were within the reference range. Furthermore, the neutrophil-to-lymphocyte ratio (NLR) and the platelet-to-lymphocyte ratio (PLR) were significantly higher in patients with low LMR (p < 0.001, Table 3).

Biochemical results indicate that patients with low LMR exhibit more marked alterations compared to those with high LMR. Alkaline phosphatase was elevated in both groups, being significantly higher in the low LMR group (p < 0.001). Serum iron and transferrin were below reference values in both groups, with significantly lower levels in the low LMR group (p = 0.013 and p = 0.001, respectively). Ferritin levels were well above the reference range in both groups, with no significant difference between them. Likewise, post-hemodialysis C-reactive protein (CRP) was within the reference range in both groups, but significantly higher in patients with low LMR (p < 0.001) (Table 4).

### Sensitivity analysis

To assess the discriminative power of LMR in classifying patients according to the median-based distribution of clinical and hematological variables, ROC analyses were performed. As shown in Fig 2, the performance of LMR varied across

**Table 1. Clinical and epidemiological characteristics of patients with chronic kidney disease (CKD) on hemodialysis.**

| Characteristic | n (%) n = 120 |
|---|---|
| **Sex** | |
| **Male** | 65 (54.2%) |
| **Female** | 55 (45.8%) |
| **Age (years)** | 58 (47, 72) |
| **Etiology of CKD** | |
| Chronic glomerulonephritis | 41 (34.2%) |
| Hypertensive nephropathy | 31 (25.8%) |
| Diabetic nephropathy | 28 (23.3%) |
| Obstructive uropathy | 5 (4.2%) |
| Polycystic kidney disease | 6 (5.0%) |
| Interstitial nephritis | 4 (3.3%) |
| Other/unknown | 5 (4.2%) |
| **Duration of hemodialysis (years)** | 7 (12.25) |
| **Duration of hemodialysis** | |
| 1–3 years | 29 (24.2%) |
| 4–6 years | 26 (21.7%) |
| >7 years | 65 (54.2%) |
| **Body Mass Index (BMI)** | |
| Underweight (<18.5) | 6 (5.4%) |
| Normal (18.5–24.9) | 73 (65.8%) |
| Overweight (25–29.9) | 25 (22.5%) |
| Obesity (≥30) | 7 (6.3%) |
| **Vascular access** | |
| Central venous catheter (CVC) | 42 (35.0%) |
| Arteriovenous fistula (AVF) | 78 (65.0%) |

[1]Median (Interquartile Range); n (%).

variables. The lowest AUC was observed for Time on Hemodialysis (AUC = 0.51, Fig 2B), indicating no better-than-random classification. The highest AUC was found in the comparison with NLR (AUC = 0.79, Fig 2E), suggesting a strong discriminatory potential. Intermediate AUC values were observed for Age (0.62, Fig 2A), Monocytes (0.74, Fig 2C), Lymphocytes (0.77, Fig 2D), and PLR (0.70, Fig 2F). The optimal LMR cutoff values ranged from 2.71 (Time on HD, Fig 2B) to 3.28 (NLR, Fig 2E), indicating that LMR values slightly above 3 may serve as a robust threshold in differentiating patient subgroups, particularly when related to systemic inflammation markers.

### Cross-sectional regression analysis

Factors associated with lower LMR values were identified through logistic regression analysis. Sociodemographic, clinical, and laboratory variables that showed significant differences in the bivariate analysis between study groups, as well as inflammation-related variables, were included. The analysis showed that risk factors associated with decreased LMR were: age (OR=1.03; p < 0.001), more than three years on hemodialysis (OR=2.17; p = 0.002), more than 7 years on hemodialysis (OR=2.38; p < 0.001), and having T2DM (OR=2.2; p = 0.001). Meanwhile, normal weight (BMI within normal range) was a risk factor (OR=2.82; p = 0.005), and female sex was identified as a protective factor (OR=0.67; p < 0.011) (Table 5).

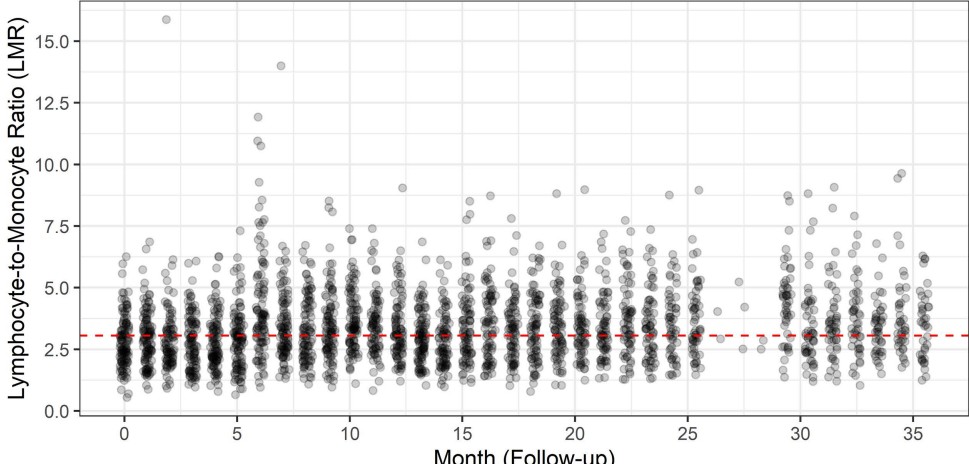

**Fig 1. Lymphocyte to Monocyte ratio from the cohort of this study.** We plotted all captured levels for LMR of these patients with a maximum follow-up time of 35.5 months. Each grey dot represents an individual patient measurement, the black dots denoting values within the central density of the data distribution for better visualization. The dashed red line marks the baseline median LMR (3.06).

## Longitudinal regression analysis

We also evaluated variables that might contribute to lower LMR using a longitudinal approach, tracking patient progress over the follow-up period (up to three years). Table 6 shows the results from fixed effects regression models. According to the obtained Beta parameters, age and hemoglobin negatively contributed to LMR levels, while arteriovenous fistula (AVF) vascular access and hematocrit showed a positive relationship with LMR.

## Discussion

The CKD population on hemodialysis evaluated in this study had a median age of 58 years and a slight male predominance. Several factors were identified as being associated with decreased LMR, considered a biomarker of inflammation: older age, more than three years on hemodialysis, presence of type 2 diabetes mellitus, and normal body weight. A direct relationship was found between LMR and hematocrit levels as well as AVF access. In contrast, female sex was a protective factor against reduced LMR.

Low LMR values are associated with an increased predisposition to developing lymphopenia and/or monocytosis. According to published evidence, this association is particularly linked to the accumulation of senescent monocytes in the context of chronic diseases such as chronic kidney disease (CKD) [30]. It is important to note that while monocyte and lymphocyte subpopulations were not directly measured in this study, the following mechanistic interpretations are based on previously published literature.

Monocyte senescence has been associated with telomere shortening, elevated intracellular levels of reactive oxygen species (ROS), and disruption of mitochondrial membrane potential [31]. Monocytes can be categorized into classical, intermediate, and non-classical subtypes. Among these, the non-classical proinflammatory monocytes exhibit the most pronounced features of senescence, followed by intermediate and then classical subsets [6].

Additionally, studies have shown that lymphopenia is related to a reduction in regulatory T cells, which are critical for modulating immune responses and preventing tissue damage mediated by inflammation [32]. Together, these immunological imbalances contribute to persistent activation of lymphocytes and macrophages, promoting the release of proinflammatory cytokines such as tumor necrosis factor-alpha (TNF-α) and interleukin-6 (IL-6), which directly contribute to glomerular injury and renal fibrosis [6].

**Table 2. Comparison of clinical characteristics in CKD patients on hemodialysis with high and low lymphocyte-to-monocyte ratio (LMR).**

| Characteristic | Low LMR (n = 61)¹ | High LMR (n = 59)¹ | p-value² |
|---|---|---|---|
| **Sex** | | | 0.5 |
| Female | 26 (43%) | 29 (49%) | |
| Male | 35 (57%) | 30 (51%) | |
| **Age (years)** | 65 (53, 73) | 54 (43, 71) | 0.01 |
| **CKD Etiology** | | | 0.15 |
| Chronic glomerulonephritis | 19 (31%) | 19 (32%) | |
| Diabetic nephropathy | 18 (30%) | 12 (20%) | |
| Hypertensive nephropathy | 15 (25%) | 16 (27%) | |
| Polycystic kidney disease | 4 (6.6%) | 3 (5.1%) | |
| Interstitial nephritis | 3 (4.9%) | 2 (3.4%) | |
| Obstructive uropathy | 2 (3.3%) | 3 (5.1%) | |
| Other/unknown | 0 (0%) | 4 (6.8%) | |
| **Diabetes status** | | | 0.1 |
| Diabetic | 18 (30%) | 10 (17%) | |
| Non-diabetic | 43 (70%) | 49 (83%) | |
| **Hemodialysis duration (years)** | 5 (2, 11) | 11 (4, 18) | 0.002 |
| **Hemodialysis time group** | | | >0.9 |
| 1–3 years | 14 (23%) | 15 (25%) | |
| 4–6 years | 14 (23%) | 12 (20%) | |
| >7 years | 33 (54%) | 32 (54%) | |
| **Vascular access** | | | 0.3 |
| Central venous catheter (CVC) | 24 (39%) | 18 (31%) | |
| Arteriovenous fistula (AVF) | 37 (61%) | 41 (69%) | |
| **Body Mass Index (BMI)** | | | 0.4 |
| Underweight (<18.5) | 2 (3.6%) | 4 (7.1%) | |
| Normal (18.5–24.9) | 40 (73%) | 33 (59%) | |
| Overweight (25–29.9) | 11 (20%) | 14 (25%) | |
| Obesity (≥30) | 2 (3.6%) | 5 (8.9%) | |

¹Median (IQR); n (%).

²Wilcoxon rank-sum test; Pearson's Chi-squared test; Fisher's exact test.

Although the use of the median is a widely adopted strategy to dichotomize continuous variables [33], this analysis provides new insights by allowing the inclusion of specific LMR cutoff values previously reported in the literature. For instance, Alsayyad et al. (2019) reported an LMR cutoff of 2.66 as an inflammatory marker predictive of diabetic nephropathy development, with a sensitivity of 44% and specificity of 92% [34]. Similarly, Ibrahim et al. (2024) identified a monocyte-to-lymphocyte ratio (MLR) cutoff of 0.3425 associated with chronic kidney disease progression in patients with micro- and macroalbuminuria [35]. While this value can be mathematically converted to an LMR of 2.92, it's crucial to acknowledge the limits of extrapolating findings between these two ratios. In addition, it is important to point out that the cutoff derived from the median in our study (3.06) remains exploratory and requires validation in independent cohort studies before it can be recommended for clinical use.

Given that the LMR or MLR may contain important information for the classification of patients with CKD undergoing hemodialysis, we have compiled associated information from a manual review of scientific literature (Table 7).

**Table 3. Comparison of hematological parameters in CKD patients on hemodialysis with high and low LMR.**

| Hematological Parameters | Low LMR (n = 61)[1] | High LMR (n = 59)[1] | Reference Range | p-value[2] |
|---|---|---|---|---|
| Leukocytes (/mm³) | 6,300 (5,350–7,595) | 6,250 (5,180–7,410) | 5,000–11,000 | 0.13 |
| Eosinophils (/mm³) | 261 (170–398) | 262 (157–457) | 0–600 | 0.6 |
| Basophils (/mm³) | 31 (20–47) | 31 (22–47) | 0–150 | 0.11 |
| Monocytes (/mm³) | 538 (433–668) | 403 (313–508) | 100–900 | <0.001 |
| Lymphocytes (/mm³) | 1,199 (977–1,469) | 1,709 (1,359–2,160) | 1,500–7,500 | <0.001 |
| Hematocrit (%) | 30.80 (29.50–32.15) | 30.70 (29.30–32.00) | 33–46 | 0.2 |
| Hemoglobin (g/dL) | 11.00 (9.70–12.15) | 11.10 (9.80–11.80) | 11–16 | 0.5 |
| Categorized Hemoglobin | | | | 0.4 |
| Hb < 11 g/dL | 536 (42%) | 509 (40%) | — | |
| Hb ≥ 11 g/dL | 738 (58%) | 752 (60%) | — | |
| Platelets (/mm³) | 189,000 (149,000–236,000) | 187,000 (144,000–232,000) | 150,000–450,000 | 0.4 |
| Neutrophil-to-lymphocyte ratio (NLR) | 3.46 (2.55–4.61) | 2.09 (1.61–2.74) | — | <0.001 |
| Platelet-to-lymphocyte ratio (PLR) | 152 (120–200) | 110 (81–142) | — | <0.001 |

**Table 4. Comparison of biochemical parameters in CKD patients on hemodialysis with high and low LMR.**

| Biochemical Parameters | Low LMR (n = 61)[1] | High LMR (n = 59)[1] | Reference Range | p-value[2] |
|---|---|---|---|---|
| Triglycerides (mg/dL) | 132 (87–175) | 134 (97–179) | 40–150 | 0.2 |
| Total Cholesterol (mg/dL) | 154 (132–178) | 162 (135–188) | 100–200 | 0.1 |
| HDL Cholesterol (mg/dL) | 37 (31–48) | 38 (32–45) | >45 | 0.9 |
| LDL Cholesterol (mg/dL) | 87 (68–106) | 96 (77–120) | 60–180 | 0.006 |
| ALT (U/L) | 15 (10–24) | 15 (10–24) | 5–32 | 0.8 |
| AST (U/L) | 17 (11–23) | 16 (12–22) | 7–33 | 0.3 |
| Alkaline Phosphatase (U/L) | 234 (153–345) | 196 (138–285) | 30–120 | <0.001 |
| Total Proteins (g/dL) | 7.00 (6.70–7.40) | 6.90 (6.60–7.30) | 6.4–8.3 | 0.11 |
| Albumin (g/dL) | 3.90 (3.70–4.10) | 4.00 (3.80–4.20) | 3.5–5.0 | <0.001 |
| C-Reactive Protein (mg/L) | 0.63 (0.29–1.72) | 0.32 (0.19–0.67) | <10 | <0.001 |
| Urea (pre-dialysis, mg/dL) | 137 (115–162) | 142 (119–164) | — | 0.022 |
| Urea (post-dialysis, mg/dL) | 32 (25–39) | 31 (25–38) | 22–55 | 0.2 |
| Creatinine (pre, mg/dL) | 9.00 (7.60–10.50) | 9.70 (8.05–11.30) | — | <0.001 |
| Creatinine (post, mg/dL) | 2.70 (2.20–3.40) | 2.70 (2.20–3.40) | 0.74–1.35 | >0.9 |
| Serum Calcium (mg/dL) | 9.00 (8.50–9.50) | 8.95 (8.50–9.40) | 8.5–10.5 | 0.6 |
| Phosphorus (mg/dL) | 4.70 (3.50–5.90) | 4.80 (3.70–5.80) | 2.9–5.0 | 0.7 |
| Parathyroid Hormone (pg/L) | 341 (254–445) | 272 (158–281) | 50–300 | 0.7 |
| Iron (µmol/L) | 9.7 (6.8–13.6) | 10.4 (8.2–14.2) | 10.7–30.4 | 0.013 |
| Transferrin (mg/dL) | 160 (136–183) | 168 (146–197) | 200–400 | 0.001 |
| Ferritin (mg/dL) | 728 (316–1353) | 633 (294–1189) | 28–365 | 0.062 |

[1]Median (IQR); n (%).

[2]Wilcoxon rank-sum test.

These results align with reports describing a direct association between age and MLR, the inverse of LMR. Clinically, both indices reflect the same phenomenon: a higher monocyte-to-lymphocyte ratio, a condition associated with chronic inflammation and the development of chronic kidney disease (CKD) [21]. Aging has been described as favoring monocyte

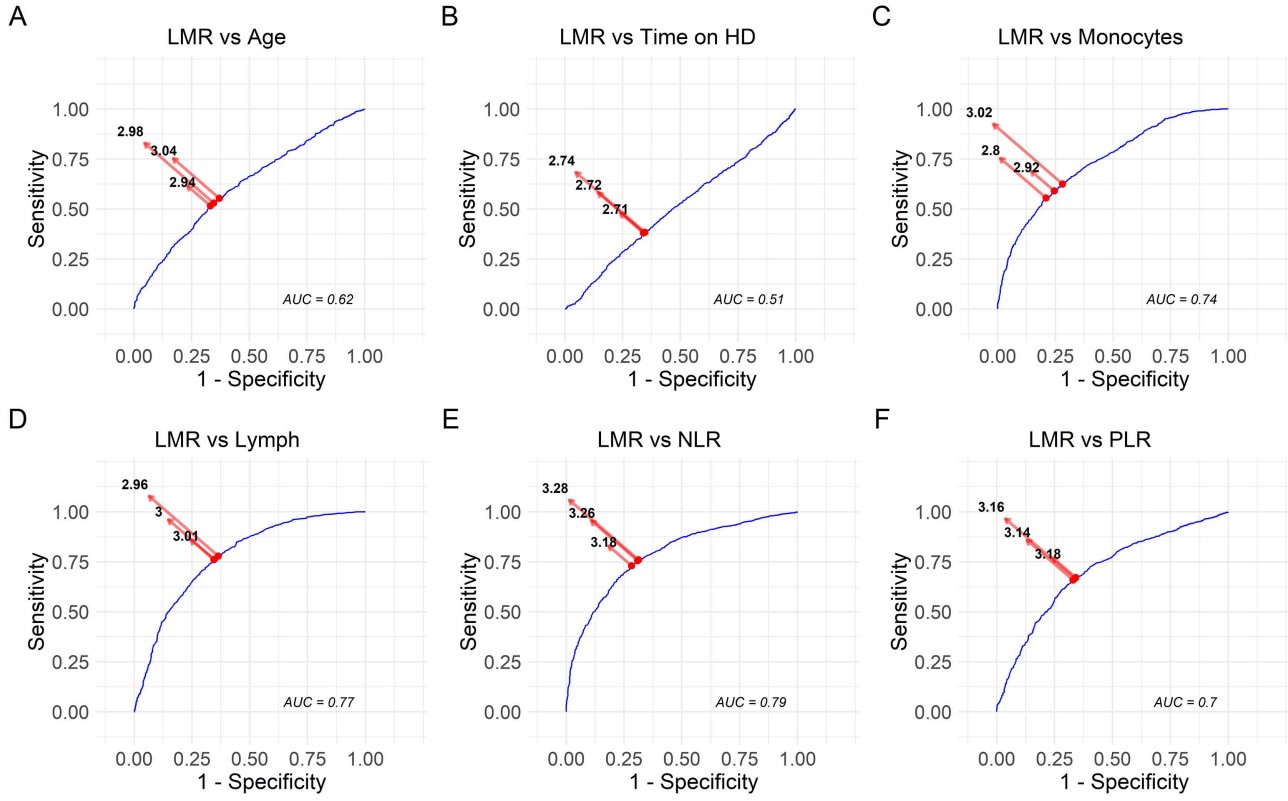

**Fig 2. Sensitivity analysis of LMR thresholds based on ROC curves for associated variables.** ROC curves depicting the performance of the lymphocyte-to-monocyte ratio (LMR) in classifying patients stratified by the median of six variables: **(A)** Age, **(B)** Time on Hemodialysis, **(C)** Monocyte count, **(D)** Lymphocyte count, **(E)** Neutrophil-to-Lymphocyte Ratio (NLR), and **(F)** Platelet-to-Lymphocyte Ratio (PLR). Each plot shows the three LMR cutoff points that yielded the highest combined sensitivity and specificity. The blue line represents the ROC curve, red dots indicate the best thresholds, and the diagonal line represents the line of no discrimination. The Area Under the Curve (AUC) is displayed in each panel.

senescence, a process accelerated in patients with end-stage CKD [6]. This phenomenon, known as immunosenescence, is intensified in environments with increased oxidative stress and elevated concentrations of pro-inflammatory cytokines in the extracellular microenvironment [6].

The observed association between longer hemodialysis duration and lower LMR is consistent with previous studies showing that CKD patients exposed for prolonged periods to a uremic environment exhibit a greater proportion of proin-flammatory intermediate monocytes, resulting in higher MLR values, the inverse of LMR [23]. In this context, Liu et al. (2024) demonstrated that elevated MLR is independently associated with increased all-cause and cardiovascular mortality in hemodialysis patients, reflecting the chronic inflammatory state characteristic of this population [9].

The lymphocyte-to-monocyte ratio (LMR) is an affordable and accessible hematological biomarker. Alsayyad et al. (2019) found that an LMR < 2.66 predicted diabetic nephropathy with high specificity (92%) [34]. Similarly, recent studies by Zhou et al. (2024) [10] and Ibrahim (2024) [35] have linked low LMR values with systemic inflammation and progression of chronic kidney disease (CKD).

Although biomarkers such as IL-6 or TNF-α offer greater immunological specificity, their routine use is limited by cost and availability. In contrast, LMR has demonstrated clinical value even in predicting cardiovascular mortality in patients undergoing hemodialysis, with prognostic performance equal to or greater than that of the neutrophil-to-lymphocyte ratio (NLR) and platelet-to-lymphocyte ratio (PLR) [27,36,37].

**Table 5. Factors associated with low lymphocyte-to-monocyte ratio (LMR) in CKD patients on hemodialysis.**

| Variable | Univariate Model | | | Multivariate Model (adjusted)* | | |
|---|---|---|---|---|---|---|
| | OR¹ | 95% CI¹ | p-value | OR¹ | 95% CI¹ | p-value |
| Age | 1.03 | 1.02–1.03 | <0.001 | 1.03 | 1.02–1.04 | <0.001 |
| Sex | | | | | | |
| Male | — | — | — | — | — | — |
| Female | 0.78 | 0.67–0.91 | 0.002 | 0.67 | 0.49–0.91 | 0.011 |
| Hemodialysis duration | | | | | | |
| 1–3 years | — | — | — | — | — | — |
| 4–6 years | 1.41 | 1.11–1.80 | 0.006 | 2.17 | 1.32–3.61 | 0.002 |
| >7 years | 1.21 | 0.98–1.49 | 0.076 | 2.38 | 1.54–3.73 | <0.001 |
| Type 2 Diabetes (DM2) | | | | | | |
| Non-diabetic | — | — | — | — | — | — |
| Diabetic | 1.66 | 1.37–2.02 | <0.001 | 2.20 | 1.40–3.50 | 0.001 |
| Body Mass Index (BMI) | | | | | | |
| Underweight (<18.5) | — | — | — | — | — | — |
| Normal (18.5–24.9) | 3.80 | 2.59–5.72 | <0.001 | 2.82 | 1.40–6.05 | 0.005 |
| Overweight (25–29.9) | 2.27 | 1.52–3.47 | <0.001 | 1.34 | 0.63–2.98 | 0.459 |
| Obesity (≥30) | 1.61 | 0.96–2.73 | 0.071 | 1.44 | 0.54–3.89 | 0.469 |

* Adjusted for: sex, age, type 2 diabetes, BMI, ferritin, and hemodialysis duration.

OR, Odds Ratio; CI, Confidence Interval.

**Table 6. Factors contributing to LMR values in CKD patients on hemodialysis.**

| Model Variable (Full Cohort) | Beta | 95% CI | p-value |
|---|---|---|---|
| Age | −0.01 | −0.02 to 0.00 | 0.024 |
| Vascular access (AVF) | 0.17 | 0.01 to 0.33 | 0.038 |
| Hematocrit (%) | 0.25 | 0.21 to 0.29 | <0.001 |
| Hemoglobin (g/dL) | −0.72 | −0.85 to −0.60 | <0.001 |
| Random intercept (patient-level) | 0.95 | | |
| Residual standard deviation | 1.00 | | |

CI, Confidence Interval; AVF, Arteriovenous Fistula; LMR, Lymphocyte-to-Monocyte Ratio.

Notably, Xiang et al. (2018) reported that elevated LMR was independently associated with increased all-cause and cardiovascular mortality in hemodialysis patients, outperforming NLR as a prognostic factor [38]. Furthermore, Ibrahim et al. (2024) found that both LMR and IL-6 positively correlated with carotid intima-media thickness, an early marker of atherosclerosis, supporting their utility in predicting disease progression in diabetic nephropathy [35].

Taken together, these findings underscore the complementary role of LMR as an inflammatory biomarker in CKD. Its inclusion in clinical assessments adds value for both routine monitoring and inflammatory risk stratification. However, some studies have questioned this association. For instance, Benck et al. found no significant correlation between dialysis duration and changes in monocyte activity, as measured by functional markers such as tissue factor or platelet-monocyte aggregates [39]. This suggests that, while relevant immune alterations occur in these patients, the relationship between dialysis duration and monocyte activation may depend on other yet unidentified factors.

**Table 7. Factors contributing to LMR values in CKD patients on hemodialysis.**

| Index Analyzed | Study Type and Pathology Studied | Key Findings Related to This Study | Reference |
|---|---|---|---|
| LMR | Prospective cohort of 3,391 non-dialysis CKD patients with up to 11.7 years of follow-up. | A higher LMR was associated with a greater risk of cardiovascular events (HR = 1.26; 95% CI: 1.16–1.36), cardiovascular death (HR = 1.27; 95% CI: 1.10–1.48), and all-cause mortality (HR = 1.18; 95% CI: 1.09–1.29). These findings support LMR as an accessible and cost-effective risk marker in CKD. | Oh et al., 2022 [22] |
| LMR | Case-control study with 100 patients with type 2 diabetes mellitus as cases and 25 controls. | The optimal LMR cutoff point for predicting diabetic nephropathy was reported as 2.66, with a sensitivity of 44% and specificity of 92%, making it a useful marker for inflammation. | Alsayyad et al., 2019 [34] |
| LMR | Retrospective cohort of 1,830 patients with type 2 diabetes mellitus (T2DM) admitted to intensive care units for the first time. | An LMR > 0.71 was associated with higher 30- and 90-day mortality. The risk of 90-day death nearly doubled (HR ≈ 1.9–2.4), confirming LMR as an independent predictor of poor prognosis. | Qiu et al., 2023 [24] |
| LMR | Retrospective longitudinal study analyzing 280 patients with diabetic foot. | Inflammatory biomarkers, such as NLR, PLR, and LMR, have shown predictive capacity for serious complications in diabetic foot patients, including peripheral arterial disease, osteomyelitis, and amputation risk. | Demirdal et al., 2018 [20] |
| LMR | Prospective cohort with 403 patients diagnosed with ST-elevation myocardial infarction (STEMI). | A decrease in LMR was found to increase the risk of long-term stroke, transient ischemic attack, non-fatal myocardial infarction, and cardiovascular mortality (p = 0.012, p = 0.001, p = 0.003, and p = 0.002, respectively). | Wang et al., 2017 [27] |
| LMR | Retrospective longitudinal study analyzing 121 patients with diffuse large B-cell lymphoma (DLBCL). | An LMR < 2 was shown to be a predictor of overall survival in DLBCL, with an HR = 2.2 (95% CI: 1.2–4.1, p = 0.011). | Beltran et al., 2019 [18] |
| LMR | Meta-analysis of 26 studies including 8,586 patients with esophageal squamous cell carcinoma (ESCC). | Low LMR, high PLR, and high NLR were associated with poor survival and a malignant phenotype, characterized by deeper tumor invasion, positive lymph node metastasis, and an advanced TNM stage in ESCC patients. | Sun et al., 2018 [16] |
| LMR | Retrospective cohort of stage III melanoma patients with microscopic metastasis in the sentinel lymph node (SLN). | NLR, LMR, and CRP provide useful prognostic information on recurrence-free survival. The combination of high CRP and low LMR had the highest predictive power for melanoma recurrence. | Schildbach et al., 2023 [8] |
| MLR | Prospective cohort of chronic kidney disease patients. | A higher MLR was a strong and independent predictor of all-cause and cardiovascular mortality, and it outperformed NLR among patients on HD. | Xiang et al., 2018 [38] |
| MLR | Cohort study of 1,809 subjects, 403 of whom had CKD. | An elevated MLR was associated with a higher risk of inflammation (OR = 2.30; 95% CI: 1.24–4.27). It had moderate predictive capacity (AUC = 0.631) with a cutoff point of 0.153, confirming it as an independent inflammatory marker in CKD. | Zhou et al., 2024 [10] |
| MLR | Prospective observational study of 90 diabetic patients. | Both MLR and IL-6 were positively correlated with carotid intima-media thickness, an early marker of atherosclerosis. Both were useful predictors of diabetic nephropathy progression. The MLR cutoff point was 0.3425, associated with CKD progression in patients with micro- and macroalbuminuria. | Ibrahim et al., 2024 [35] |
| MLR | Retrospective observational study with 771 patients with acute kidney injury (AKI) secondary to an acute hemorrhagic stroke. | In patients with acute hemorrhagic stroke, MLR was strongly associated with AKI risk (OR = 8.27; AUC = 0.73; cutoff point = 0.5556). It was also related to in-hospital mortality (OR = 3.13; AUC = 0.62; cutoff point = 0.7059). | Jiang et al., 2022 [29] |
| MLR | Cross-sectional study with 65 CKD patients. | MLR not only showed a positive correlation with disease progression and risk of death but also outperformed other indicators like NLR, TNF-α, CRP, and estimated GFR. | Sari et al., 2024 [23] |
| MLR | Retrospective cohort with 11,262 eligible patients, 3,015 of whom had CKD. | An elevated MLR is independently associated with higher all-cause and cardiovascular mortality in hemodialysis patients. High MLR significantly increased mortality (adjusted HR up to 1.99). Survival was lower in the elevated MLR groups (log-rank p < 0.001). | Liu et al., 2024 [9] |
| MLR | Prospective cohort of 1,280 participants from China followed for nearly 2 years. | 2.55% developed new CKD. An elevated MLR was associated with a higher risk of new CKD (crude HR = 16.12; adjusted HR = 8.89), confirming it as an independent predictor. These findings support the value of MLR in the early detection and prevention of CKD. | Zhang et al., 2020 [21] |

Although our study found an association between T2DM and lower LMR values, this link has not been widely documented. Nonetheless, prior studies have reported that T2DM is associated with a chronic inflammatory state triggered by sustained or transient hyperglycemia, promoting an increase in inflammatory monocytes, which may help explain the reduced LMR observed [12,32,40]. Similar alterations have been described in other chronic inflammatory diseases, such as peripheral artery disease and various cardiovascular conditions [19,27]. In this context, LMR may serve as a more sensitive indirect marker of systemic inflammation than monocyte count alone. Moreover, a combination of elevated monocyte count and reduced lymphocyte count has been associated with increased cardiovascular events and mortality in this population [24].

Regarding BMI, 65.8% of patients in this study had a normal BMI, while approximately 30% were overweight or obese. This distribution contrasts with national studies, such as those provided by the Peruvian Health Minister – MINSA (2015), which reported overweight prevalence between 35.1% and 47.8% and obesity between 17.5% and 30.2% [41].

When analyzing the relationship between BMI and LMR, it was observed that patients with low LMR more frequently had a BMI within the normal range compared to those with high LMR (73% vs. 59%). However, this difference did not reach statistical significance when assessing the association between LMR and overweight or obesity. While many studies recognize obesity as a risk factor for chronic inflammatory diseases, its specific association with LMR remains unclear and requires further research [42,43].

Some studies have proposed an "obesity paradox," where higher BMI may be associated with better outcomes in certain clinical contexts. In this sense, Wang (2019) reported that elevated BMI could act as a protective factor in patients with advanced chronic inflammatory diseases such as CKD on hemodialysis [44].

In our study, both groups of patients showed iron deficiency and reduced transferrin levels, with significantly lower levels in the low LMR group. Serum ferritin was elevated in both groups, with a more pronounced trend in the low LMR group, although not statistically significant. These findings are consistent with the chronic inflammatory context, where ferritin acts as an acute-phase reactant. Inflammation was positively associated with elevated ferritin levels and negatively with the presence of anemia, a condition linked to increased risk of clinical complications and premature mortality in this population [45].

Renal anemia, associated with hypoxia, inflammation, and oxidative stress in CKD, may contribute to decreased LMR by promoting an increase in proinflammatory monocytes and/or a reduction in lymphocytes, as previously described by Badura et al. (2024) [46]. However, our study revealed an inverse pattern: hemoglobin levels were negatively correlated with LMR. This may reflect reactive lymphocytosis during early immune responses or acute infections, which could temporarily coexist with anemia, as observed in our cohort.

The use of AVF in hemodialysis patients has been associated with a lower incidence of infectious complications and reduced immune activation compared to central venous catheters [47]. This reduced inflammatory burden may explain the observed association between AVF use and higher LMR values in this study.

Patients with low LMR exhibited significantly higher levels of alkaline phosphatase (ALP) and CRP, suggesting a more pronounced inflammatory state. ALP, traditionally used as a marker of bone metabolism, has recently been recognized as an indicator of systemic inflammation and cardiovascular risk in CKD patients. Recent studies have shown that elevated ALP levels are associated with CKD progression and increased mortality, independent of other factors such as bone metabolism or liver function [48].

CRP is a widely used acute-phase reactant and inflammation marker. Elevated CRP levels have been correlated with decreased LMR, reflecting an immune imbalance characterized by relative lymphopenia and monocytosis. Recent studies have identified that the combination of high CRP and low LMR is associated with worse clinical outcomes in various conditions, including chronic inflammatory diseases and neoplasms [8]. These findings support the utility of LMR as an accessible and low-cost biomarker for assessing inflammatory burden in CKD patients. Its correlation with established markers such as ALP and CRP reinforces its potential in risk stratification and disease monitoring.

Female sex was identified as a protective factor against reduced LMR. A similar finding was reported by Buttle et al. (2021), who observed lower MLR, LMR's inverse, in women with tuberculosis, another chronic inflammatory disease [49]. This difference may be influenced by the modulatory effects of sex hormones, particularly estrogens, which, through interaction with the alpha receptor, have been shown to reduce IL-6 expression in human monocytes [49].

Current immunological markers for early diagnosis and monitoring of end-stage CKD are often costly and have limited availability, restricting their use in routine clinical practice. In this context, our study highlights the potential of LMR as an accessible and low-cost inflammatory biomarker derived from the complete blood count. Its application could facilitate continuous monitoring of patients with chronic diseases such as CKD; a condition closely linked to high morbidity and mortality.

It is worth noting that other inflammatory biomarkers such as neutrophil-to-lymphocyte ratio (NLR), platelet-to-lymphocyte ratio (PLR), and LMR have shown predictive value for severe complications in patients with diabetic foot, including peripheral artery disease, osteomyelitis, and risk of amputation [20]. Moreover, available evidence supports that MLR, the inverse of LMR, is a more consistent and predictive inflammatory marker than NLR and PLR in various clinical contexts. In patients with acute hemorrhagic stroke, MLR was significantly associated with in-hospital mortality, while NLR showed limited predictive power [29]. In CKD, MLR was not only positively correlated with inflammation, disease progression, and mortality risk, but also outperformed other indicators such as NLR, TNF-α, CRP, and estimated glomerular filtration rate (eGFR) [9,10,23].

The findings of this study reinforce the clinical value of the LMR as an indicator of the inflammatory state in patients with chronic kidney disease. This suggests that longitudinal monitoring of LMR could serve as a simple and accessible tool for the dynamic stratification of risk in hemodialysis. Serial assessments would enable the identification of patients with persistent or progressive inflammation, facilitate the prediction of cardiovascular complications and mortality, and guide targeted interventions aimed at optimizing dialysis adequacy, improving anemia management, and preventing infections. The routine integration of LMR into clinical practice could therefore promote more personalized decision-making and the early identification of patients at higher risk.

Despite their contribution to their field, this study has some limitations. It includes restricted access to detailed information on patients' prior comorbidities, as well as the inability to characterize specific monocyte and lymphocyte subtypes. Nonetheless, the data obtained and analyzed allow us to conclude that the LMR is a useful and accessible inflammatory biomarker for monitoring disease severity in Peruvian CKD patients.

Moreover, dialysis adequacy (Kt/V or URR) is a relevant confounding factor, as it may influence both inflammatory markers and clinical outcomes. In our cohort, these data were not available and thus could not be included in the multivariate adjustment, which represents a limitation of the study. However, in a prior pilot study conducted in our population, the average Kt/V was 1.7 ± 0.2 (https://n9.cl/mv2k2, S2 File), well above the minimum recommended threshold of 1.3 [50], which is widely recognized as an indicator of effective hemodialysis. This finding reduces the likelihood that inadequate clearance influenced inflammatory levels or outcomes related to the LMR.

## Conclusion

Our findings support the value of LMR as an accessible, low-cost, and clinically relevant biomarker for assessing inflammatory status in hemodialysis patients with CKD. Its associations with immunological and biochemical parameters, including ferritin, CRP, and ALP, as well as with demographic and clinical features such as sex and diabetes mellitus, reinforce its potential as a monitoring and risk stratification tool. Given its availability from routine complete blood counts, LMR could be incorporated into clinical practice as a complementary marker in the comprehensive evaluation of CKD patients. Further longitudinal studies with larger sample sizes and more detailed immunophenotypic characterization are required to validate its prognostic value and explore its applicability across other populations and clinical settings.

## Supporting information

**S1 File.** Database used in this study. The spreadsheet includes two main pages. The first one includes data loaded into R, and the second is the dictionary of variables.
(XLSX)

**S2 File.** Pilot study presented as scientific poster. Results presented at the 2019 Congress of the Latin American Society of Nephrology and Hypertension (https://slanh.net/). Content available in Spanish.
(PDF)

## Acknowledgments

We sincerely thank the staff of Clinical CENESA S.A. for their essential support and collaboration, which were instrumental to the completion of this study. We are especially grateful to Dr. César Liendo Liendo and Lic. Elva Fuertes Yalán for their invaluable assistance and unwavering commitment in providing all the necessary resources to achieve the objectives of this research.

## Author contributions

**Conceptualization:** Daysi Zulema Diaz-Obregón, Víctor Arrunátegui Correa.

**Data curation:** Gabriela Goyoneche Linares, Alexis G. Murillo Carrasco.

**Formal analysis:** Daysi Zulema Diaz-Obregón, Gabriela Goyoneche Linares, Alexis G. Murillo Carrasco.

**Investigation:** Daysi Zulema Diaz-Obregón, Gabriela Goyoneche Linares, Ana Granda Alacote, Michael Bryant Castro Núñez, Andrea Vanessa Llanos Diaz, Víctor Arrunátegui Correa.

**Methodology:** Daysi Zulema Diaz-Obregón, Gabriela Goyoneche Linares, Alexis G. Murillo Carrasco, Víctor Arrunátegui Correa.

**Resources:** Ana Granda Alacote, Michael Bryant Castro Núñez, Andrea Vanessa Llanos Diaz, Katherine Susan Rufasto Goche, Antonio Mauricio Sánchez Cotrina.

**Supervision:** Víctor Arrunátegui Correa.

**Validation:** Daysi Zulema Diaz-Obregón, Gabriela Goyoneche Linares, Alexis G. Murillo Carrasco, Joel de León Delgado.

**Visualization:** Daysi Zulema Diaz-Obregón, Ana Granda Alacote, Michael Bryant Castro Núñez, Andrea Vanessa Llanos Diaz, Katherine Susan Rufasto Goche, Antonio Mauricio Sánchez Cotrina, Joel de León Delgado.

**Writing – original draft:** Daysi Zulema Diaz-Obregón, Víctor Arrunátegui Correa.

**Writing – review & editing:** Daysi Zulema Diaz-Obregón, Alexis G. Murillo Carrasco, Michael Bryant Castro Núñez, Andrea Vanessa Llanos Diaz, Katherine Susan Rufasto Goche, Antonio Mauricio Sánchez Cotrina, Joel de León Delgado.

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
