## [Decision Letter · Decision Letter 0]

7 Jul 2025

Dear Dr. Carrasco,

We look forward to receiving your revised manuscript.

Kind regards,

Tatsuo Shimosawa, M.D., Ph.D.

Academic Editor

PLOS ONE

Journal Requirements:

2. We note that your Data Availability Statement is currently as follows: All relevant data are within the manuscript and its Supporting Information files

Reviewers' comments:

Reviewer's Responses to Questions

**Comments to the Author**

1. Is the manuscript technically sound, and do the data support the conclusions?

Reviewer #1: Yes

Reviewer #2: Yes

2. Has the statistical analysis been performed appropriately and rigorously?

Reviewer #1: Yes

Reviewer #2: Yes

3. Have the authors made all data underlying the findings in their manuscript fully available?

Reviewer #1: Yes

Reviewer #2: Yes

4. Is the manuscript presented in an intelligible fashion and written in standard English?

Reviewer #1: Yes

Reviewer #2: Yes

Reviewer #1: General Comments

This retrospective cohort study explores the variability of the lymphocyte-to-monocyte ratio (LMR) in patients with chronic kidney disease (CKD) on hemodialysis and its association with demographic, clinical, and laboratory parameters. The research is timely and relevant, considering the growing interest in inexpensive, accessible inflammatory markers in CKD populations. The study is well-structured and statistically sound, utilizing both cross-sectional and longitudinal regression analyses. The findings are consistent with current knowledge and emphasize the potential role of LMR in inflammation monitoring. However, while the study is methodologically adequate, several aspects need clarification or expansion. In particular, further contextualization of LMR among other inflammatory markers (e.g., NLR, PLR, IL-6, TNF-α) would improve the discussion. Additionally, the use of a single population-based cut-off value for LMR (3.06) could benefit from further justification or sensitivity analysis. Overall, the manuscript presents valuable findings but would benefit from major revision before it is suitable for publication.

1 While the use of LMR as an inflammatory marker is well-established, the manuscript would benefit from explicitly stating how this study adds to existing literature.

2 The authors used the median LMR value (3.06) as a binary cut-off. While methodologically acceptable in exploratory studies, this lacks external validation.

3 The models adjust for major variables such as age, sex, BMI, and diabetes. However, was dialysis adequacy (Kt/V or URR) evaluated?

4 The discussion includes detailed speculation about classical/intermediate/non-classical monocytes. However, the study does not perform monocyte subtyping. Please clarify that this discussion is based on existing literature and not directly measured in the current dataset.

Reviewer #2: This study investigates whether the lymphocyte-to-monocyte ratio (LMR), derived from standard hematological parameters, could serve as a useful inflammatory marker in patients undergoing hemodialysis. Although the topic is clinically relevant and potentially insightful, some important issues remain unresolved.

Major Points

1) The authors used the median LMR value (3.06) as a cutoff to categorize the hemodialysis patients in this study. It would be helpful to support the decision with citations from previous studies that share a similar focus or clinical relevance.

2) The manuscript should provide detailed statistics (e.g., mean and distribution) for both the high- and low-LMR groups to allow for a better interpretation of the findings.

3) The observation that the type of vascular access may affect the LMR is interesting. Were there any clinical background differences (e.g., age, comorbidities) between the arteriovenous fistula (AVF) and non-AVF groups that could explain the differences in LMR?

4) The association between LMR and sex was noteworthy. Please clarify whether age differences existed between the sexes, whether age influenced the LMR, and whether this sex difference was also present in older individuals, such as postmenopausal women.

5) Hemoglobin (Hb) and hematocrit (Ht) levels exhibited different trends in relation to LMR. The authors should explore the possible biological or methodological reasons for this discrepancy.

6) The manuscript frequently cites studies investigating the monocyte-to-lymphocyte ratio (MLR), although the present study focused on the LMR instead. As these two markers are mathematically inverse and can differ in clinical interpretation, the authors should clarify whether the findings from MLR studies can be extrapolated to LMR. Additionally, the rationale for selecting LMR over MLR in this study should be discussed.

7) Several references cited as supporting LMR actually discuss MLR (e.g., references 8, 9, 15, 18, 28, 29, and 43). To ensure clarity, please specify whether each reference pertains to LMR or MLR.

Minor Points

1) Table 2: The high-LMR group included 59 patients, but the number of cases listed under CKD etiology did not add up to this total. Please review and revise the case counts and their respective percentages accordingly.

2) Table 4: The use of "TGP" and "TGO" is unnecessary as these are non-standard terms. It is sufficient to report liver enzymes as "AST" and "ALT.” In addition, because "Uric Acid" was not measured, it should be removed from the table.

3) Page 6, Lines 26–29: Reference 15 relates to MLR and CKD onset, not aging. References 6 and 16 also did not discuss LMR specifically. These references should be reviewed and, if necessary, replaced with more appropriate ones.

4) Page 16, Lines 23–31: Reference 44 by Valga et al. (2020) focused on erythropoietin resistance and did not support the conclusions regarding PLR/NLR as predictors of mortality or cardiovascular events. This reference should be revised or removed.

**Do you want your identity to be public for this peer review?** For information about this choice, including consent withdrawal, please see our Privacy Policy

Reviewer #1: No

Reviewer #2: No

---

## [Author Response · Author response to Decision Letter 1]

1 Aug 2025

Editor comments

Comment E.1

Two experts raised serious concerns related with the integrity of the current study. They require extensive revision.

Response E.1

Dear Editor Tatsuo Shimosawa, M.D., Ph.D., thank you for allowing us to revise our manuscript. In this version, we have replied to all reviewers’ comments and modified the main text and analyses of our study accordingly. Please find all responses below.

Journal Requirements

Comment J.1

Response J.1

Dear PlosOne team, we have adjusted our manuscript to comply with the journal’s style requirements. Please check the new version of our submission.

Comment J.2

We note that your Data Availability Statement is currently as follows: All relevant data are within the manuscript and its Supporting Information files

Response J.2

Thank you for the comment. In this new version, we attached the minimal dataset as Supplementary File 1.

Reviewers comments

Reviewer 1

Comment 1.1

General Comments

This retrospective cohort study explores the variability of the lymphocyte-to-monocyte ratio (LMR) in patients with chronic kidney disease (CKD) on hemodialysis and its association with demographic, clinical, and laboratory parameters. The research is timely and relevant, considering the growing interest in inexpensive, accessible inflammatory markers in CKD populations. The study is well-structured and statistically sound, utilizing both cross-sectional and longitudinal regression analyses. The findings are consistent with current knowledge and emphasize the potential role of LMR in inflammation monitoring.

Response 1.1

Thank you for this sincere and kind appreciation of our manuscript. In this new version, we have revised all your following comments. We hope to have properly replied to your concerns.

Comment 1.2

However, while the study is methodologically adequate, several aspects need clarification or expansion. In particular, further contextualization of LMR among other inflammatory markers (e.g., NLR, PLR, IL-6, TNF-α) would improve the discussion.

Response 1.2

Thank you for this relevant comment. In the new version of our manuscript, we have improved the discussion section. Regarding the contextualization of LMR among other inflammatory markers, we included this paragraph:

‘The lymphocyte-to-monocyte ratio (LMR) is an affordable and accessible hematological biomarker. Alsayyad et al. (2019) found that an LMR < 2.66 predicted diabetic nephropathy with high specificity (92%). Similarly, recent studies by Zhou et al. (2024) and Ibrahim (2024) have linked low LMR values with systemic inflammation and progression of chronic kidney disease (CKD).

Although biomarkers such as IL-6 or TNF-α offer greater immunological specificity, their routine use is limited by cost and availability. In contrast, LMR has demonstrated clinical value even in predicting cardiovascular mortality in patients undergoing hemodialysis, with prognostic performance equal to or greater than that of the neutrophil-to-lymphocyte ratio (NLR) and platelet-to-lymphocyte ratio (PLR) (Catabay et al., 2017; Sevecan et al., 2019).

Notably, Xiang et al. (2018) reported that elevated LMR was independently associated with increased all-cause and cardiovascular mortality in hemodialysis patients, outperforming NLR as a prognostic factor. Furthermore, Ibrahim et al. (2024) found that both LMR and IL-6 positively correlated with carotid intima-media thickness, an early marker of atherosclerosis, supporting their utility in predicting disease progression in diabetic nephropathy.

Taken together, these findings underscore the complementary role of LMR as an inflammatory biomarker in CKD. Its inclusion in clinical assessments adds value for both routine monitoring and inflammatory risk stratification. ‘

Comment 1.3

Additionally, the use of a single population-based cut-off value for LMR (3.06) could benefit from further justification or sensitivity analysis. Overall, the manuscript presents valuable findings but would benefit from major revision before it is suitable for publication.

Response 1.3

Thank you for your valuable observation. In our study, we used a lymphocyte-to-monocyte ratio (LMR) cutoff of 3.06, corresponding to the median value of our study population. This approach is considered valid in exploratory studies when no previously validated threshold is available.

In response to your suggestion, we conducted an additional sensitivity analysis using ROC curve methodology. The results indicated a range of optimal LMR cutoff values between 2.71 and 3.28, which includes the median-derived value of 3.06. Based on these findings, we have included a “Sensitivity Analysis” section in the Methods and Results, and added the following paragraph to the Discussion:

"Although the use of the median is a widely adopted strategy to dichotomize continuous variables (Hui-Ju et al., 2021), this analysis provides new insights by allowing the inclusion of specific LMR cutoff values previously reported in the literature. For instance, Alsayyad et al. (2019) reported an LMR cutoff of 2.66 as an inflammatory marker predictive of diabetic nephropathy development, with a sensitivity of 44% and specificity of 92%. Similarly, Ibrahim et al. (2024) identified a monocyte-to-lymphocyte ratio (MLR) cutoff of 0.3425 associated with chronic kidney disease progression in patients with micro- and macroalbuminuria, which translates to an LMR of 2.92."

Comment 1.4

1. While the use of LMR as an inflammatory marker is well-established, the manuscript would benefit from explicitly stating how this study adds to existing literature.

Response 1.4

Although the lymphocyte-to-monocyte ratio (LMR) is recognized as a useful inflammatory marker, our study provides novel evidence by analyzing its longitudinal variability in patients with chronic kidney disease (CKD) undergoing hemodialysis. This approach enables assessment of its dynamic behavior over time. Additionally, while using a median-based cutoff value of 3.06, we conducted a sensitivity analysis incorporating alternative thresholds reported in the literature, which yielded consistent results.

We also expanded the discussion by comparing LMR with other inflammatory markers, including the neutrophil-to-lymphocyte ratio (NLR), platelet-to-lymphocyte ratio (PLR), interleukin-6 (IL-6), and tumor necrosis factor-alpha (TNF-α). Previous studies, such as those by Xiang et al. (2018) and Ibrahim et al. (2024), have shown that LMR may be a more robust predictor of mortality and disease progression than other biomarkers, reinforcing its clinical utility and accessibility in resource-limited settings.

Comment 1.5

2. The authors used the median LMR value (3.06) as a binary cut-off. While methodologically acceptable in exploratory studies, this lacks external validation.

Response 1.5

Thank you for this relevant comment. To strengthen our findings, we have incorporated a sensitivity analysis as commented on in our response 1.3. Unfortunately, we did not identify any publicly available external datasets with complete longitudinal data suitable for additional validation. However, our group is currently compiling a longitudinal dataset of CKD patients undergoing hemodialysis. We expect to complete its organization soon and make it publicly accessible.

Comment 1.6

3. The models adjust for major variables such as age, sex, BMI, and diabetes. However, was dialysis adequacy (Kt/V or URR) evaluated?

Response 1.6

Thank you for your valuable observation. Indeed, dialysis adequacy is a clinically relevant factor that may influence inflammatory markers and mortality outcomes. Before the current study, we conducted a pilot study in which the average Kt/V in our population was 1.7 ± 0.2. It was well above the minimum recommended threshold of 1.3, which is widely recognized as an indicator of effective hemodialysis treatment. This reduces the likelihood that inadequate clearance influenced inflammatory levels or outcomes related to the LMR (Pérez-García, 2019). Although these results were not included in the present manuscript, they were presented at the 2019 Congress of the Latin American Society of Nephrology and Hypertension (https://slanh.net/). The poster can be accessed here: https://drive.google.com/file/d/1kePxIymcNBXF5G3stwTMin_dmVsxJEd-/view?usp=sharing.

References

● Pérez-García R, Jaldo M, Alcázar R, de Sequera P, Albalate M, Puerta M, et al. El Kt/V alto, a diferencia del Kt, se asocia a mayor mortalidad: importancia de la V baja. Nefrología. 2019;39(1):58–66.

Comment 1.7

4. The discussion includes detailed speculation about classical/intermediate/non-classical monocytes. However, the study does not perform monocyte subtyping. Please clarify that this discussion is based on existing literature and not directly measured in the current dataset.

Response 1.7

Thank you for this comment. In this new version, we have proofread our discussion to make it less speculative. Please see the corrected text here:

‘Low LMR values are associated with an increased predisposition to developing lymphopenia and/or monocytosis. According to published evidence, this association is particularly linked to the accumulation of senescent monocytes in the context of chronic diseases such as chronic kidney disease (CKD) (26). Monocyte senescence has been associated with telomere shortening, elevated intracellular levels of reactive oxygen species (ROS), and disruption of mitochondrial membrane potential (Zhou et al., 2021). Monocytes can be categorized into classical, intermediate, and non-classical subtypes. Among these, the non-classical proinflammatory monocytes exhibit the most pronounced features of senescence, followed by intermediate and then classical subsets (6).

Additionally, studies have shown that lymphopenia is related to a reduction in regulatory T cells, which are critical for modulating immune responses and preventing tissue damage mediated by inflammation (27). Together, these immunological imbalances contribute to persistent activation of lymphocytes and macrophages, promoting the release of proinflammatory cytokines such as tumor necrosis factor-alpha (TNF-α) and interleukin-6 (IL-6), which directly contribute to glomerular injury and renal fibrosis (6).’

Reviewer 2

Comment 2.1

This study investigates whether the lymphocyte-to-monocyte ratio (LMR), derived from standard hematological parameters, could serve as a useful inflammatory marker in patients undergoing hemodialysis. Although the topic is clinically relevant and potentially insightful, some important issues remain unresolved.

Response 2.1

Thank you for this sincere and kind appreciation of our manuscript. In this new version, we have revised all your following comments. We hope to have properly replied to your concerns.

Comment 2.2

Major Points

1. The authors used the median LMR value (3.06) as a cutoff to categorize the hemodialysis patients in this study. It would be helpful to support the decision with citations from previous studies that share a similar focus or clinical relevance.

Response 2.2

Thank you for your valuable observation. In our study, we used a lymphocyte-to-monocyte ratio (LMR) cutoff of 3.06, corresponding to the median value of our study population. This approach is considered valid in exploratory studies when no previously validated threshold is available.

In response to your suggestion, we conducted an additional sensitivity analysis using ROC curve methodology. The results indicated a range of optimal LMR cutoff values between 2.71 and 3.28, which includes the median-derived value of 3.06. Based on these findings, we have included a “Sensitivity Analysis” section in the Methods and Results, and added the following paragraph to the Discussion:

"Although the use of the median is a widely adopted strategy to dichotomize continuous variables (Hui-Ju et al., 2021), this analysis provides new insights by allowing the inclusion of specific LMR cutoff values previously reported in the literature. For instance, Alsayyad et al. (2019) reported an LMR cutoff of 2.66 as an inflammatory marker predictive of diabetic nephropathy development, with a sensitivity of 44% and specificity of 92%. Similarly, Ibrahim et al. (2024) identified a monocyte-to-lymphocyte ratio (MLR) cutoff of 0.3425 associated with chronic kidney disease progression in patients with micro- and macroalbuminuria, which translates to an LMR of 2.92."

Comment 2.3

2. The manuscript should provide detailed statistics (e.g., mean and distribution) for both the high- and low-LMR groups to allow for a better interpretation of the findings.

Response 2.3

Thank you for this comment. We have generated a new Figure 1 to display the full range of LMR values for all patients in this study.

Comment 2.4

3. The observation that the type of vascular access may affect the LMR is interesting. Were there any clinical background differences (e.g., age, comorbidities) between the arteriovenous fistula (AVF) and non-AVF groups that could explain the differences in LMR?

Response 2.4

Thank you for this important comment. In response to your comment, we looked at potential differences related to Vascular Access with Age (p=0.06), Sex (p=0.5), Etiology (p=0.4), Diabetes mellitus 2 (p>0.9), and Comorbidity (p=0.4).

Comment 2.5

4. The association between LMR and sex was noteworthy. Please clarify whether age differences existed between the sexes, whether age influenced the LMR, and whether this sex difference was also present in older individuals, such as postmenopausal women.

Response 2.5

Thank you for this insightful comment. In the revised version, we clarified that neither LMR (p = 0.7) nor sex (p = 0.8) was influenced by age in our cohort. Additionally, we assessed the association between age and LMR categories specifically in patients over 45 years old and found no significant relationship (p = 0.6).

Comment 2.6

5. Hemoglobin (Hb) and hematocrit (Ht) levels exhibited different trends in relation to LMR. The authors should explore the possible biological or methodological reasons for this discrepancy.

Response 2.6

We thank the reviewer for this insightful observation. In response, we have added a discussion of this point in the revised manuscript (Discussion section). Specifically, while both hemoglobin and hematocrit are typically correlated as indicators of red blood cell mass, our longitudinal analysis revealed a positive association between hematocrit and LMR, and a negative association between hemoglobin and LMR.

This apparent discrepancy may reflect distinct physiological or inflammatory dynamics. One plausible explanation is the presence of reactive lymphocytosis during early immune responses or subclinical infections, which may transiently coexist with anemia—a condition commonly observed in CKD patients. Thus, despite lower hemoglobin levels, an increase in lymphocyte count could elevate the LMR. Conversely, hematocrit, being more stable and less sensitive to short-term fluctuations, may better reflect the underlying chronic inflammatory burden. This differentiation highlights the complex immuno-hematologic landscape in hemodialysis patients and underscores the need for future studies with more refined immunophenotyping to fully elucidate these patterns.

Comment 2.7

6. The manuscript frequently cites studies investigating the monocyte-to-lymphocyte ratio (MLR), although the present study focused on the LMR instead. As these two markers are mat

---

## [Decision Letter · Decision Letter 1]

8 Aug 2025

Dear Dr. Carrasco,

Thank you for submitting your manuscript to PLOS ONE. After careful consideration, we feel that it has merit but does not fully meet PLOS ONE’s publication criteria as it currently stands. Therefore, we invite you to submit a revised version of the manuscript that addresses the points raised during the review process.

We look forward to receiving your revised manuscript.

Kind regards,

Tatsuo Shimosawa, M.D., Ph.D.

Academic Editor

PLOS ONE

Journal Requirements:

Reviewer's Responses to Questions

**Comments to the Author**

Reviewer #1: All comments have been addressed

Reviewer #2: All comments have been addressed

2. Is the manuscript technically sound, and do the data support the conclusions?

Reviewer #1: Yes

Reviewer #2: Yes

3. Has the statistical analysis been performed appropriately and rigorously?

Reviewer #1: Yes

Reviewer #2: Yes

4. Have the authors made all data underlying the findings in their manuscript fully available?

Reviewer #1: Yes

Reviewer #2: Yes

5. Is the manuscript presented in an intelligible fashion and written in standard English?

Reviewer #1: Yes

Reviewer #2: Yes

Reviewer #1: The authors have responded to most of my initial comments in a constructive and detailed manner. I appreciate the substantial improvements, including the addition of a sensitivity analysis for the LMR cut-off, expanded discussion comparing LMR with other inflammatory markers, and corrections to tables and terminology. The manuscript is now clearer and more informative. However, several important issues remain insufficiently addressed. These points should be resolved before the manuscript can be considered ready for publication.

1. While this study focuses on LMR, many cited studies investigated the monocyte-to-lymphocyte ratio (MLR). Although the authors note that LMR and MLR are mathematically related, their clinical interpretation and predictive ability are not always interchangeable. Please explicitly indicate in the text (or ideally in a summary table) whether each cited reference pertains to LMR or MLR. Clarify the limits of extrapolating findings from MLR to LMR and avoid implying full equivalence in prognostic implications.

2. The addition of ROC-based sensitivity analysis is commendable. However, the analysis was performed within the same dataset, limiting external generalizability. Please explicitly state in the Discussion that the median-derived cut-off (3.06) remains exploratory and requires validation in independent cohorts before it can be recommended for clinical use.

3. The adequacy of dialysis (Kt/V, URR) is an important confounder that may influence inflammatory markers and outcomes. The authors refer to a pilot study, but no adequacy data for the current study population are presented. If available, please provide actual Kt/V or URR values for the study cohort and consider including them in multivariate models. If not available, explicitly acknowledge this as a limitation.

4. The section on monocyte subpopulations remains largely speculative, as no direct measurements were performed. Please clearly state in the Discussion that these mechanistic interpretations are based on literature, not on data from the current study.

5. The study’s main novelty is the longitudinal evaluation of LMR variability. However, the practical implications for patient management and prognosis remain underdeveloped. Please expand on how monitoring longitudinal LMR might alter clinical decision-making or improve risk stratification in hemodialysis patients.

Reviewer #2: The author has responded carefully and appropriately to the previous reviewers' comments.

An additional analysis using the ROC curve to explore the relationship between LMR and other inflammatory markers (such as NLR and PLR) is a valuable addition to the study.

While the content and structure show improvement, I still have significant concerns regarding the accuracy and appropriateness of references.

Please consider the following points for further revision:

1) New Figure 1: The figure uses both gray and black dots. Please add an explanation in the figure caption to clarify what each type of dot represents.

2) Page 5, line 5: Reference 7 has been changed from the previous version to the current version. Please verify its appropriateness and reconsider whether reference 6 is still necessary in this context.

3) Page 5, lines 31-32: Please re-check reference 23, as it does not appear to discuss CKD progression.

4) Page 6, line 3: Please confirm that reference 10 is appropriate, as it does not primarily focus on peritoneal dialysis patients.

5) Page 15, line 6: Please confirm that reference 21 is a suitable citation, as it does not include hemodialysis patients.

**Do you want your identity to be public for this peer review?** For information about this choice, including consent withdrawal, please see our Privacy Policy

Reviewer #1: No

Reviewer #2: No

---

## [Author Response · Author response to Decision Letter 2]

5 Sep 2025

Editor comments

Comment E.1

There still remain some minor pionts to be rivised.

Response E.1

Dear Editor Tatsuo Shimosawa, M.D., Ph.D., thank you for allowing us to revise our manuscript. In this version, we have replied to all reviewers’ comments and modified the main text and analyses of our study accordingly. Please find all responses below.

Journal Requirements

Comment J.1

Response J.1

Dear PlosOne team, Thank you for this advice. In this version, we have revised all suggestions to include and revise references to make proper decisions.

Comment J.2

Response J.2

Thank you for the comment. In this new version, we have revised all references and changed some of them. All changes are highlighted in the track_changes file.

Reviewer 1

Comment 1.1

The authors have responded to most of my initial comments in a constructive and detailed manner. I appreciate the substantial improvements, including the addition of a sensitivity analysis for the LMR cut-off, expanded discussion comparing LMR with other inflammatory markers, and corrections to tables and terminology. The manuscript is now clearer and more informative. However, several important issues remain insufficiently addressed. These points should be resolved before the manuscript can be considered ready for publication.

Response 1.1

Thank you for the comment. In this new version, we have included improvements to attend to your comments. We hope our answers are aligned with your requirements.

Comment 1.2

While this study focuses on LMR, many cited studies investigated the monocyte-to-lymphocyte ratio (MLR). Although the authors note that LMR and MLR are mathematically related, their clinical interpretation and predictive ability are not always interchangeable. Please explicitly indicate in the text (or ideally in a summary table) whether each cited reference pertains to LMR or MLR. Clarify the limits of extrapolating findings from MLR to LMR and avoid implying full equivalence in prognostic implications.

Response 1.2

Thank you for the comment. In this new version, we modified lines 390-395 to comment on the limitations of extrapolating data based on LMR or MLR. In addition, we summarised all relevant information for MLR or LMR in Table 7.

Comment 1.3

The addition of ROC-based sensitivity analysis is commendable. However, the analysis was performed within the same dataset, limiting external generalizability. Please explicitly state in the Discussion that the median-derived cut-off (3.06) remains exploratory and requires validation in independent cohorts before it can be recommended for clinical use.

Response 1.3

Thank you for the comment. In this new version, we added the following text: ‘In addition, it is important to point out that the cutoff derived from the median in our study (3.06) remains exploratory and requires validation in independent cohort studies before it can be recommended for clinical use.’ (page 15)

Comment 1.4

The adequacy of dialysis (Kt/V, URR) is an important confounder that may influence inflammatory markers and outcomes. The authors refer to a pilot study, but no adequacy data for the current study population are presented. If available, please provide actual Kt/V or URR values for the study cohort and consider including them in multivariate models. If not available, explicitly acknowledge this as a limitation.

Response 1.4

Thank you for the comment. In this new version, we added the following paragraph on page 19:

‘Moreover, dialysis adequacy (Kt/V or URR) is a relevant confounding factor, as it may influence both inflammatory markers and clinical outcomes. In our cohort, these data were not available and thus could not be included in the multivariate adjustment, which represents a limitation of the study. However, in a prior pilot study conducted in our population, the average Kt/V was 1.7 ± 0.2 (https://n9.cl/mv2k2, S2 File), well above the minimum recommended threshold of 1.3 (Pérez-García, 2019), which is widely recognized as an indicator of effective hemodialysis. This finding reduces the likelihood that inadequate clearance influenced inflammatory levels or outcomes related to the LMR.’

Comment 1.5

The section on monocyte subpopulations remains largely speculative, as no direct measurements were performed. Please clearly state in the Discussion that these mechanistic interpretations are based on literature, not on data from the current study.

Response 1.5

Thank you for the comment. In this new version, we added the following text between lines 368-371:

‘It is important to note that while monocyte and lymphocyte subpopulations were not directly measured in this study, the following mechanistic interpretations are based on previously published literature.’

Comment 1.6

The study’s main novelty is the longitudinal evaluation of LMR variability. However, the practical implications for patient management and prognosis remain underdeveloped. Please expand on how monitoring longitudinal LMR might alter clinical decision-making or improve risk stratification in hemodialysis patients.

Response 1.6

Thank you for the comment. In this new version, we added the following text on page 19:

‘The findings of this study reinforce the clinical value of the LMR as an indicator of the inflammatory state in patients with chronic kidney disease. This suggests that longitudinal monitoring of LMR could serve as a simple and accessible tool for the dynamic stratification of risk in hemodialysis. Serial assessments would enable the identification of patients with persistent or progressive inflammation, facilitate the prediction of cardiovascular complications and mortality, and guide targeted interventions aimed at optimizing dialysis adequacy, improving anemia management, and preventing infections. The routine integration of LMR into clinical practice could therefore promote more personalized decision-making and the early identification of patients at higher risk.’

Reviewer 2

Comment 2.1

The author has responded carefully and appropriately to the previous reviewers' comments. An additional analysis using the ROC curve to explore the relationship between LMR and other inflammatory markers (such as NLR and PLR) is a valuable addition to the study. While the content and structure show improvement, I still have significant concerns regarding the accuracy and appropriateness of references. Please consider the following points for further revision:

Response 2.1

Thank you for the comment. In this new version, we have included improvements to attend to your comments. We hope our answers are aligned with your requirements.

Comment 2.2

New Figure 1: The figure uses both gray and black dots. Please add an explanation in the figure caption to clarify what each type of dot represents.

Response 2.2

Thank you for the comment. In this new version, we complemented the legend of Figure 1 to clarify the meaning of these different grey/black tones. They are related to data density.

‘Fig 1. Lymphocyte to Monocyte ratio from the cohort of this study. We plotted all captured levels for LMR of these patients with a maximum follow-up time of 35.5 months. Each grey dot represents an individual patient measurement, the black dots denoting values within the central density of the data distribution for better visualization. The dashed red line marks the baseline median LMR (3.06).’

Comment 2.3

Page 5, line 5: Reference 7 has been changed from the previous version to the current version. Please verify its appropriateness and reconsider whether reference 6 is still necessary in this context.

Response 2.3

Thank you for the comment. We realized an involuntary mistake, now been corrected.

References 6-7:

6. Ong S-M, Hadadi E, Dang T-M, Yeap W-H, Tan CT-Y, Ng T-P, et al. The pro-inflammatory phenotype of the human non-classical monocyte subset is attributed to senescence. Cell Death Dis. 2018;9: 266. doi:10.1038/s41419-018-0327-1

7. Xia Y, Yang Q, Wu SY, Wu Z, Li Q, Du J. Interferon lambda modulates proinflammatory cytokines production in PBMCs from patients with chronic kidney disease. Hum Immunol. 2023;84: 464–470. doi:10.1016/j.humimm.2023.06.001

Comment 2.4

Page 5, lines 31-32: Please re-check reference 23, as it does not appear to discuss CKD progression.

Response 2.4

Thank you for the comment. We apologize for this mistake. We corrected this reference for Sari et al., 2024.

Sari D, Wardhani P, Puspitasari Y, Suryantoro S. Association of monocyte-to-lymphocyte ratio, neutrophil-to-lymphocyte ratio, and tumor necrosis factor-α in various stages of chronic kidney disease. Journal of Advanced Biotechnology and Experimental Therapeutics. 2024;7: 346. doi:10.5455/jabet.2024.d29

Comment 2.5

Page 6, line 3: Please confirm that reference 10 is appropriate, as it does not primarily focus on peritoneal dialysis patients.

Response 2.5

Thank you for the comment. We updated the description of that reference to align with their content.

‘’Zhou et al. (2024) demonstrated that MLR is a simple and low-cost biomarker that reflects systemic inflammation and helps physicians in the management of nephropathies [10]. ‘

Comment 2.6

Page 15, line 6: Please confirm that reference 21 is a suitable citation, as it does not include hemodialysis patients.

Response 2.6

Thank you for the comment. Indeed, the referenced study corresponds to a prospective cohort of 1,280 participants in China, followed for nearly two years, in which the variability of the MLR and its association with the development of chronic kidney disease (CKD) were analyzed.

The change is made to: “Conversely, elevated MLR values, reflecting lower LMR, have been linked to aging [21], increased cardiovascular risk [22], CKD progression [10,23], and high mortality due to CKD [22].”

---

## [Decision Letter · Decision Letter 2]

15 Sep 2025

Variability of the lymphocyte-to-monocyte ratio in patients with chronic kidney disease on hemodialysis

PONE-D-25-31524R2

Dear Dr. Carrasco,

We’re pleased to inform you that your manuscript has been judged scientifically suitable for publication and will be formally accepted for publication once it meets all outstanding technical requirements.

Kind regards,

Tatsuo Shimosawa, M.D., Ph.D.

Academic Editor

PLOS ONE

Additional Editor Comments (optional):

Reviewer #1:

Reviewer #2:

Reviewers' comments:

Reviewer's Responses to Questions

**Comments to the Author**

Reviewer #1: All comments have been addressed

Reviewer #2: All comments have been addressed

2. Is the manuscript technically sound, and do the data support the conclusions?

Reviewer #1: Yes

Reviewer #2: Yes

3. Has the statistical analysis been performed appropriately and rigorously?

Reviewer #1: Yes

Reviewer #2: Yes

4. Have the authors made all data underlying the findings in their manuscript fully available?

Reviewer #1: Yes

Reviewer #2: Yes

5. Is the manuscript presented in an intelligible fashion and written in standard English?

Reviewer #1: Yes

Reviewer #2: Yes

Reviewer #1: The authors have substantially revised the manuscript in response to the previous round of comments, and I commend their thorough and constructive efforts.

The following key improvements are noted:

1. The manuscript now explicitly clarifies the difference between LMR and MLR, with a summary table indicating which references pertain to each. This avoids conflating the two indices and improves interpretability.

2. The text now appropriately acknowledges that the ROC-derived cut-off (3.06) is exploratory, derived from the study dataset, and requires validation in independent cohorts.

3. Although Kt/V and URR data were not available for the current cohort, the authors clearly acknowledge this limitation and provide supporting information from a prior pilot study.

4. The authors have expanded the discussion to highlight the potential clinical utility of longitudinal LMR monitoring in risk stratification, prediction of cardiovascular events, and guiding management strategies in hemodialysis patients.

5. In addition, the reference list and figure captions have been carefully revised, and overall clarity of the manuscript has improved.

The manuscript now satisfactorily addresses all major concerns raised in the prior review. While the absence of dialysis adequacy data remains a limitation, it is transparently acknowledged, and does not invalidate the main findings. The study provides novel and valuable insights into LMR variability in hemodialysis patients. I therefore consider the manuscript acceptable for publication in its current form.

Reviewer #2: The revised manuscript accurately addresses all of my comments. I can also confirm that the references have been corrected appropriately. I have no further comments.

**Do you want your identity to be public for this peer review?** For information about this choice, including consent withdrawal, please see our Privacy Policy

Reviewer #1: No

Reviewer #2: No

---

## [Editor Report · Acceptance letter]

PONE-D-25-31524R2

PLOS ONE

Dear Dr. Carrasco,

I'm pleased to inform you that your manuscript has been deemed suitable for publication in PLOS ONE. Congratulations! Your manuscript is now being handed over to our production team.

Kind regards,

on behalf of

Prof. Tatsuo Shimosawa

Academic Editor

PLOS ONE